# Lower Extremity Arterial Disease in Type 2 Diabetes Mellitus: Metformin Inhibits Femoral Artery Ultrastructural Alterations as well as Vascular Tissue Levels of AGEs/ET-1 Axis-Mediated Inflammation and Modulation of Vascular iNOS and eNOS Expression

**DOI:** 10.3390/biomedicines11020361

**Published:** 2023-01-26

**Authors:** Ayed A. Shati, Amro Maarouf, Amal F. Dawood, Nervana M. Bayoumy, Youssef A. Alqahtani, Refaat A. Eid, Saeed M. Alqahtani, Mohamed Abd Ellatif, Bahjat Al-Ani, Alia Albawardi

**Affiliations:** 1Department of Child Health, College of Medicine, King Khalid University, Abha 61421, Saudi Arabia; 2Department of Clinical Biochemistry, Birmingham Heartlands Hospital, University Hospitals Birmingham NHS Foundation Trust, Birmingham B9 5SS, UK; 3Department of Basic Medical Sciences, College of Medicine, Princess Nourah Bint Abdulrahman University, P.O. Box 84428, Riyadh 11671, Saudi Arabia; 4Department of Physiology, College of Medicine, King Saud University, Riyadh 11461, Saudi Arabia; 5Department of Pathology, College of Medicine, King Khalid University, Abha 61421, Saudi Arabia; 6Department of Surgery, College of Medicine, King Khalid University, Abha 61421, Saudi Arabia; 7Department of Clinical Biochemistry, College of Medicine, King Khalid University, Abha 61421, Saudi Arabia; 8Department of Medical Biochemistry, College of Medicine, Mansoura University, Mansoura 35516, Egypt; 9Department of Physiology, College of Medicine, King Khalid University, Abha 61421, Saudi Arabia; 10Department of Pathology, College of Medicine and Health Sciences, United Arab Emirates University, Al Ain 15551, United Arab Emirates

**Keywords:** femoral artery, diabetes, AGEs, ET-1, NOS, inflammation, dyslipidemia, metformin

## Abstract

Lower extremity arterial disease (LEAD) is a major risk factor for amputation in diabetic patients. The advanced glycation end products (AGEs)/endothelin-1 (ET-1)/nitric oxide synthase (NOS) axis-mediated femoral artery injury with and without metformin has not been previously investigated. Type 2 diabetes mellitus (T2DM) was established in rats, with another group of rats treated for two weeks with 200 mg/kg metformin, before being induced with T2DM. The latter cohort were continued on metformin until they were sacrificed at week 12. Femoral artery injury was established in the diabetic group as demonstrated by substantial alterations to the femoral artery ultrastructure, which importantly were ameliorated by metformin. In addition, diabetes caused a significant (*p* < 0.0001) upregulation of vascular tissue levels of AGEs, ET-1, and iNOS, as well as high blood levels of glycated haemoglobin, TNF-α, and dyslipidemia. All of these parameters were also significantly inhibited by metformin. Moreover, metformin treatment augmented arterial eNOS expression which had been inhibited by diabetes progression. Furthermore, a significant correlation was observed between femoral artery endothelial tissue damage and glycemia, AGEs, ET-1, TNF-α, and dyslipidemia. Thus, in a rat model of T2DM-induced LEAD, an association between femoral artery tissue damage and the AGEs/ET-1/inflammation/NOS/dyslipidemia axis was demonstrated, with metformin treatment demonstrating beneficial vascular protective effects.

## 1. Introduction

Central obesity, which forms the basis of insulin resistance and the metabolic syndrome, results in the production of pro-inflammatory cytokines such as tumor necrosis factor alpha (TNF-α) and interleukin-6 (IL-6). Under physiological circumstances, these cytokines are suppressed by the anti-inflammatory and anti-fibrotic effects of adiponectin through the stimulation of the anti-inflammatory cytokine, interleukin-10 (IL-10). In obese subjects, there is impaired secretion of adiponectin and leptin, resulting in a chronic inflammatory state and insulin resistance [1].

T2DM is a chronic metabolic disease that manifests from insulin resistance and/or production and is closely associated with metabolic syndrome, which generally affects the elderly and overweight middle-aged populations [2,3]. Moreover, it is strongly associated with cardiovascular disease (CVD) including endothelial dysfunction with up to 80% of diabetic patients developing CVD, and an estimated more than two-thirds of patients with T2DM die of cardiovascular complications [4,5], thus putting significant pressures upon healthcare systems. Additionally, an association between cardiovascular complications secondary to diabetes and the increase in glycated hemoglobin (marker for diabetes) was reported in a study that followed up diabetic middle-aged subjects for 17 years [4].

In lower extremity arterial disease (LEAD), atherosclerosis induced by dyslipidemia and diabetes mellitus causes occlusive disease of lower extremity arteries, thereby increasing the risk of amputation [6]. To highlight this financial burden, it is estimated that the cost of caring for patients with LEAD and diabetes in the USA to be $380 billion annually [6].

The association between inflammation and diabetes is well documented in humans and animal studies, as demonstrated by the following: (i) upregulation of biomarkers of inflammation such as C-reactive protein, leukocytosis, and interleukin-6 in diabetes (with the amelioration of these biomarkers in subjects treated with antidiabetic drugs) [7]; (ii) the expression of tumor necrosis factor-alpha (TNF-α) being increased in adipose tissue of obesity and diabetic animal models [8]; and (iii) TNF-α inhibited the phosphorylation of amino acid tyrosine of insulin receptor and its substrates by insulin [9]. Diabetes also increases vascular levels of advanced glycation end products (AGEs) and endothelin-1 (ET-1) [10]. AGEs induces vascular injury upon binding its receptor via augmenting tissue inflammation and oxidative stress [11,12]. Indeed, high levels of AGEs are associated with atherosclerosis [13], coronary artery disease [14], and hypertension [15]. ET-1 is a peptide made and secreted by the vascular endothelial cells and is recognized as a very potent endogenous vasoconstrictor, thus inducing vascular dysfunction [16]. In addition, both ET-1 and TNF-α inhibit endothelial NOS (eNOS) enzyme that participate in endothelial dysfunction [17,18].

Metformin has pleotropic (both anti-diabetic and anti-inflammatory) effects and is reported to inhibit diabetes-induced dyslipidaemia, atherosclerosis development [19,20] as well as reducing the incidence of heart failure and heart failure mortality in patients via the augmentation of AMP-activated protein kinase and NO bioavailability, and inhibition of AGEs deposition in myocardium and blood vessels [21]. Moreover in recent years, metformin has shown promising extra-antihyperglycemic characteristics such as anti-proliferative and pro-apoptotic effects in various cancer models allowing cancer patients to reduce their dose of chemotherapeutic agents [22].

However, the association between femoral artery injury and vascular expression of AGEs, ET-1, and NOS (as well as inflammation and dyslipidemia with and without metformin incorporation in animal models of T2DM) has not been previously investigated. Thus, the study aim herein was to assess this specific association as well as potential for amelioration in diabetic rats that were being treated with metformin.

## 2. Materials and Methods

### 2.1. Animals

Wistar male rats weighing 170–200 g were maintained in a clean facility with a 12-h light/dark cycle. They had unrestricted access to water and food. Under the project license number H-01-R-059 authorized by Princess Nourah Bint Abdulrahman University research ethical committee; we adhered to the Guide for the Care and Use of Laboratory Animals that was published by the US National Institutes of Health (NIH publication No. 85-23, revised 1996).

### 2.2. Experimental Design

Following an acclimatization period, twenty-four rats were randomly distributed, in equal numbers, into three groups. These being, the control non-treated group (control): rats fed for 12 weeks with standard laboratory chow. The next group were, the T2DM group (model) in which diabetes was induced using standard methods [23]; a combination of a high carbohydrate and fat diet (HCFD) and a streptozotocin injection at day 14. These rats were kept on a HCFD until the end of the experiment, at week 12. HCFD contained 10.1% fat, 10% fructose, 4% cholesterol, 17% protein, 51.6% carbohydrates as well as mineral and vitamin mixture (Collino et al., British Journal of Pharmacology 2010, 160:1892–1902). The protective group (Met + T2DM), in a similar manner to the model group, were induced to develop T2DM, but in addition, this group of rats were given a daily dose of metformin (200 mg/kg) from day one until the end of the experiment, on week 12. Blood was withdrawn at the end of the experiment, just before the rats were culled by cervical dislocation and blood vessels tissue samples were harvested.

### 2.3. Determination of Vascular and Blood Levels of AGEs, ET-1, TNF-α, TG, CHOL, LDL-C, HDL-C, HbA1c, and Glucose

AGEs and ET-1 femoral artery levels were assessed in tissue homogenates using ELISA kits provided by Ray Biotech, USA, and Abcam, Cambridge, UK, respectively, according to manufacturer’s instructions. Blood levels of TNF-α (ELISA kit BIOTANG INC, Cat. No. R6365, Lexington, MA, USA), HbA1c (ELISA kit Crystal Chem, Inc., Elk Grove Village, IL, USA), TG, CHOL, LDL-C, and HDL-C (HUMAN Diagnostics, Wiesbaden, Germany) were used according to the manufacturer’s instructions. Blood glucose was assessed using a Randox reagent kit (Sigma-Aldrich, Randox Laboratories, Antrim, UK) according to the manufacturer’s instructions.

### 2.4. Detection of eNOS and iNOS mRNAs by Quantitative Real Time Polymerase Chain Reaction (qRT-PCR)

Total RNA was isolated from femoral arteries harvested from all rats using Trizol Reagent (Qiagen, Valanceia, CA, USA) and reverse-transcribed according to the manufacturer’s instructions with the Fermentas cDNA synthesis kit (Fermentas, MA, USA). The primers specific for inducible NOS (iNOS) (sense 5′-CACCACCCTCCTTGTTCAAC-3′, and antisense 5′-CAATCCACAACTCGCTCCAA-3), eNOS (sense 5′-TATTTGATGCTCGGGACTGC-3′, and antisense 5′-AAGATTGCCTCGGTTTGTTG-3′) and -actin in Master Mix containing SYBR-Green Supermix (Molecular Probe, Eugene, OR, USA) were used to amplify cDNA samples. Relative gene expression levels were assessed using the comparative Ct technique.

### 2.5. Transmission Electron Microscopy

Small pieces of the harvested femoral arteries were fixed at 4 °C in 4% glutaraldehyde (SERVA, Frankfurt, Germany) with 0.2 M cacodylate buffers (TAAB essential for microscopy, Aldermaston, Berks, UK) and processed for TEM as mentioned before [24]. Contrasted ultrathin sections with uranyl acetate and lead citrate were examined and photographed using a Philips EM 208S TEM (FEI Company, Eindhoven, The Netherlands) at accelerating voltage of 80 Kv.

### 2.6. Statistical Analysis

The data was expressed as mean ± standard deviation (SD). Data was processed and analyzed using the SPSS version 10.0 (SPSS, Inc., Chicago, IL., USA). One-way ANOVA was carried out followed by Tukey’s post hoc test. Pearson correlation statistical analysis was determined for detection of a probable significance between two different parameters. Results were considered significant if *p* ≤ 0.05.

## 3. Results

### 3.1. Induction of Lower Extremity Arterial Disease (LEAD) Secondary to Diabetes

In diabetic patients, femoral artery lesions have been reported to be the second most pronounced extremity arterial lesions, after the crural artery [25]. Therefore, we tested the hypothesis that diabetes can induce femoral artery ultrastructural alterations. T2DM was developed in the experimental group, and transmission electron microscopy examinations of femoral artery as well as blood levels of glucose and animal body weight were assessed 10 weeks post diabetic induction (Figure 1). Representative TEM image of a muscular artery section from control rats (A); lined by endothelial cell (En), endothelial cell nucleus (N) and intact plasma membrane (arrowhead), forming the intima of the blood vessel (arrow). In addition, intact external elastic lamina (e), smooth muscle cells (SMC), and vascular lumen (Lu) were identified. Whereas, a section of a muscular artery from diabetic rats (B) demonstrates apoptotic endothelial cell (En), pyknotic nucleus (N), and surface membrane blebbing (arrowhead). The intimal lining of the blood vessel is attenuated and disrupted (arrow). The external elastic lamina is fragmented (e), and pleomorphic smooth cells (SMC) and vascular lumen (Lu) are identified. Quantitative analysis of 50 fields in each group demonstrated a profound endothelial cell (C) and external elastic lamina (D) injuries in the diabetic group. A significant (*p* < 0.0001) increase in blood glucose (E) and decrease in body weight (F) were observed in the diabetic group compared to the control rats.

### 3.2. Metformin Inhibits Vascular and Blood Levels of AGEs, ET-1, iNOS, TNF-α, as well as Dyslipidemia and Glycemia Induced by Diabetes

In cell signalling, ET-1 is located downstream of AGEs [26] and upstream of TNF-α [27], and in chronic heart failure, rats with higher serum levels of ET-1 and TNF-α have reduced heart function and survival [28]. Therefore, we assessed in our rat model of diabetes-induced LEAD femoral artery tissue, levels of AGEs, ET-1, iNOS, and eNOS, and blood levels of TNF-α and biomarkers of dyslipidemia (TG, CHOL, LDL-C, and HDL-C) and glycemia (glucose and HbA1c) with and without metformin treatment (Figure 2 and Table 1). Diabetes augmented levels of AGEs (A), ET-1 (B), TNF-α (C), iNOS (D,E), which appeared to be significantly (*p* ≤ 0.001) but not completely inhibited by metformin. Whereas, metformin augmented eNOS femoral artery levels that had been deleteriously affected by diabetes (D,F). 

The data presented in Table 1 shows metformin protecting against diabetes-modulated glucose, HbA1c, TG, CHOL, LDL-C, and HDL-C. 

### 3.3. Metformin Is Associated with the Inhibition of Diabetes-Induced LEAD

Based on the above data showing that diabetes had induced femoral artery ultrastructural damage and that metformin appeared to inhibit femoral artery tissue levels of AGEs, ET-1, and iNOS, we then sought to determine whether metformin treatment could inhibit diabetes-induced LEAD. Representative TEM images of the femoral artery layers, tunica intima (endothelial layer) and tunica media (smooth muscle layer) obtained from the control group (Figure 3A,B), the model group (Figure 3C,D), and metformin-treated group (Figure 3E,F) are depicted. Sections of a muscular artery from control rats demonstrate the tunica intima (A) endothelial cell (En), endothelial cell nucleus (N), and plasma membrane (arrowhead) forming the intimal surface of the blood vessel (arrow). Intact external elastic lamina (e), smooth muscle cells (SMC), and vascular lumen (Lu) are also shown. Whereas, TEM imaging of the tunica media (B) demonstrate normal smooth muscle cells (SMC) enclosed by intact plasma membranes (arrow), a single nucleus (N), few mitochondria (m), rough endoplasmic reticulum and lattice-like networks of actin and myosin filaments (asterisks). 

TEM image representing femoral artery sections of diabetic rats displayed in (C) show a muscular artery section with damaged apoptotic endothelial cell (En), pleomorphic nucleus (N) and disrupted plasma membrane (arrowhead), resting on an attenuated damaged intimal surface (arrow). In addition, fragmented external elastic lamina (e), pleomorphic smooth cells (SMC), and vascular lumen (Lu) are shown. Whereas, a TEM image representing a section of tunica media of a muscular artery from diabetic rats (**D**) show apoptotic smooth muscle cell (SMC), nuclei with peripheral crescents of compacted chromatin, enclosed by irregular-shaped plasma membranes (arrow), and damage to the actin and myosin of the lattice-like network (asterisk) and nuclei (N). 

Metformin treatment showed substantial protection of the femoral artery ultrastructure in tunica intima (E) as demonstrated by improved endothelial cell (En) morphology, including the nucleus (N) and plasma membrane (arrowhead), forming the intimal surface (arrow). Intact external elastic lamina (e) was also noted with few vacuoles (V) being seen. Metformin also appeared to effectively protect the tunica media (F) as demonstrated by enhancement of the smooth muscle cell (SMC) morphology, in the form of an intact plasma membrane (arrow) and a single nucleus (N), few mitochondria (m), rough endoplasmic reticulum and lattice-like networks of actin and myosin filaments (asterisk). Few vacuoles (V) are also seen. Additionally, quantification of endothelial layer damage derived from TEM analysis, showed an effective inhibition (*p* < 0.0001) of tunica intima injury by metformin (G).

### 3.4. Correlation between Score of Tunica Intima Injury and AGEs/ET-1/TNF-α/NOS Axis and Biomarkers of Glycemia and Dyslipidemia

In finding an association between the pathogenesis of diabetes-induced LEAD and the AGEs/ET-1/TNF-α/NOS axis**,** we evaluated the correlation between the score of tunica intima damage and the tissue and blood levels of AGEs, ET-1, TNF-α, NOS, TG, HDL-C, and HbA1c (Figure 4). This also would suggest that metformin can be effective in treating severe diabetic complications such as LEAD. A significant (*p* < 0.0001) correlation between the tunica intima injury score and the following parameters; AGEs (r= 0.955) (A), ET-1 (r = 0.885) (B), TNF-α (r = 0.938) (C), iNOS (r = 0.956) (D), eNOS (r = −0.853) (E), HbA1c (r = 0.734) (F), TG (r = 0.877) (G), and HDL-C (r = −0.813) (H) were observed.

## 4. Discussion

Endothelial dysfunction is the critical process in the development of diabetic microangiopathies and cardiovascular disease. Several complex mechanisms and signal molecules such as, increased reactive oxygen species (ROS) production, generation of advanced glycation end-products (AGEs) and, uncoupling of eNOS, will synergistically promote endothelium inflammation, apoptosis and impaired endothelial response [29] as well as associated systemic dysmetabolic milieu, low- grade inflammation and oxidative stress [30]. Preventing endothelial dysfunction is therefore critical in attenuating the risk of CVD development.

Metformin is known to exert vascular atheroprotection by ameliorating many of the maladaptive processes described above. This is in addition to its well- recognized insulin-sparing and sensitizing action, which is inherent in obesity and T2DM. The resulting beneficial effects on the arterial wall, both on endothelial and smooth muscle cells, may protect vasculature from fibrosis and remodeling, independent of the glucose lowering effect [30]. 

To the best of our knowledge, this report is the first to investigate, in a diabetes-induced lower extremity arterial disease (LEAD) animal model: (i) femoral artery expression of three cell signalling molecules (the advanced glycation end products (AGEs), endothelin-1 (ET-1), and NOS enzymes involved in vascular pathophysiology [16,26]; (ii) femoral artery ultrastructural alterations; and (iii) metformin’s amelioration of the above mentioned parameters. It has been reported that up to 40% of patients with T2DM have increased femoral artery intima- media thickness with 22.3% having definite atherosclerotic plaque disease that can be measured by ultrasonography [31]. Therefore, we mimicked LEAD in rats secondary to T2DM induction (Figure 1). Furthermore, metformin treatment was associated with the suppression of AGEs, ET-1, TNF-α, iNOS, dyslipidemia, as well as femoral artery ultrastructural damage that had been induced by diabetes (Figure 5). 

Furthermore, a significant correlation between femoral artery tunica intima injury and AGEs/ET-1/TNF-α/NOS axis, dyslipidemia, and glycemia was also demonstrated. Thus, these findings were congruent with our working hypothesis that diabetes-induced LEAD can cause femoral artery injuries associated with the upregulation of vascular AEGs, ET-1, iNOS, inflammation, as well as dyslipidemia that can importantly be ameliorated by the anti-hyperglycemic drug, metformin. This further corroborates our recent reports of the various beneficial effects of metformin that metformin; insofar that it is associated with the inhibition of aortic AGEs and ultrastructural damage induced by diabetes [24] as well as its inhibition of renal artery expression of AGEs, ET-1, and iNOS in diabetic rats [10]. Metformin has also been shown to inhibit AGEs-induced proinflammatory cytokines such as TNF-α in murine macrophages [32]. Moreover treating individuals with polycystic ovary syndrome for 6 months with metformin has been shown to significantly reduce plasma levels of ET-1 [33]. These reports appear to be commensurate with our findings which show significant inhibition of AGEs and ET-1 femoral artery levels in those treated with metformin. 

Upregulation of vascular tissue levels of AGEs within the aorta and renal artery after T2DM induction in rats was associated with the development of aortopathy, diabetic nephropathy, systemic hypertension, kidney injury as well as fibrosis [10,24]. Human studies have also suggested an association of AGEs and its receptor, AGER/RAGE with coronary artery disease in T2DM patients [14], diabetes-induced ischemic heart disease [34], and arterial stiffness in type 1 diabetic patients [15]. These studies would appear to mirror the principal findings of our study. Moreover, previous reports on vascular ultrastructural alterations would appear to agree with our own findings, specifically; (i) ultrastructural alterations in peripheral arteries is observed in diabetic patients with ischaemic lower limbs [35]; (ii) ultrastructural damage in myocardial cells occurs secondary to T2DM in rats [36]; (iii) ultrastructural alterations occur in the glomeruli of diabetic patients [37]; and (vi) ultrastructural alterations in certain blood components obtained from patients with T2DM participate in the pathogenesis of cardiovascular disease secondary to diabetes [38] In addition, the pathophysiological mechanisms of LEAD in diabetic patients is previously summarized by the involvement of glycemia/AGEs/inflammation/ET-1/vasoconstriction/endothelial dysfunction axis-mediated LEAD associated with dyslipidemia and hypertension [39,40] supporting our findings of diabetes-induced LEAD in a rat model of the disease (Figure 5). 

This study demonstrates that in a rat model of diabetes-induced LEAD, the induction of femoral artery ultrastructural alterations and expression of vascular AGEs, ET-1, and iNOS as well as dyslipidemia and inflammation, appeared to be inhibited by metformin for a period of 12 weeks. This study would provide further justification for the routine use of the antidiabetic drug metformin in patients with type 2 diabetes mellitus, regardless of their glycemic control.

### Limitations of the Study

In addition to the evaluation of basic metabolic parameters (glucose, HbA1c and lipid profile), as well as transmission electron microscopy examination of femoral artery specimens, the utility of doppler ultrasound assessment of the femoral artery would be additionally helpful in evaluating arterial diameter and blood flow/velocity in each group, thereby providing more strength to the observed findings.

## Figures and Tables

**Figure 1 biomedicines-11-00361-f001:**
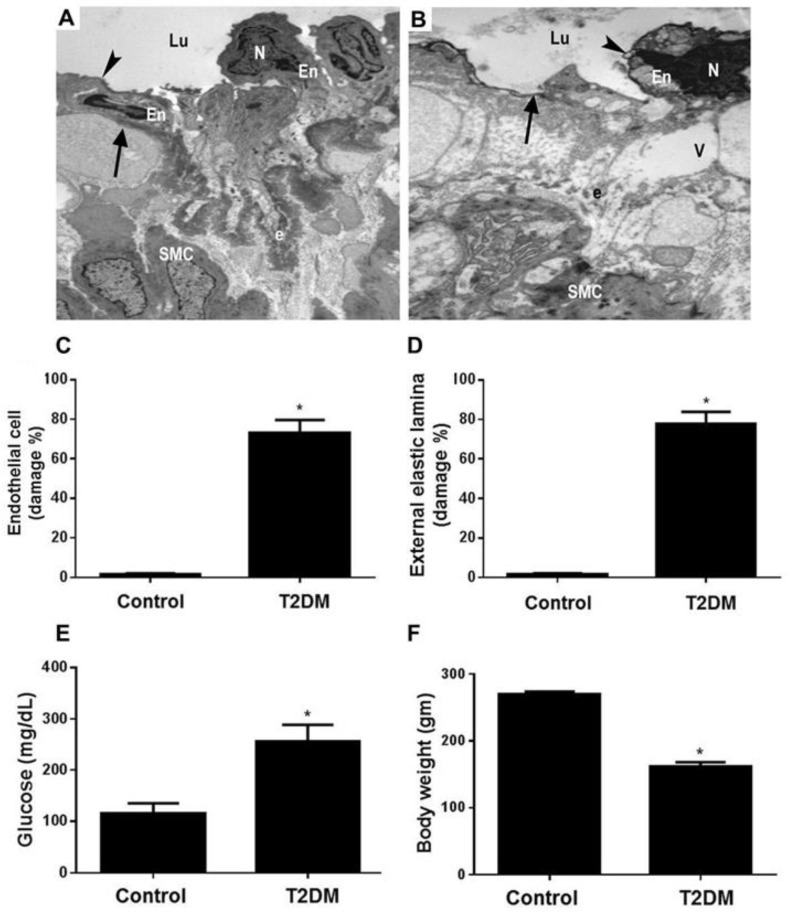
Induction of diabetes-induced LEAD in rats. TEM images (×6000) of femoral artery sections obtained at the end of the experiment of the control group (**A**) and the diabetic group (**B**) are illustrated. Note that the arrowhead in (**A**) points to the intact plasma membrane, and in (**B**) points to the surface membrane blebbing. The arrow in (**A**) points to the intima of the blood vessel, and in (**B**) points to the disrupted intimal lining of the blood vessel. N: nucleus; Lu: vascular lumen; En: endothelial cell; e: external elastic lamina; V: vacuole; SMC: smooth muscle cell. The histogram in (**C**) represents a quantitative analysis of % endothelial cells damage in femoral artery sections from the control and diabetic groups. Whereas, histogram in (**D**) represents a quantitative analysis of % external elastic lamina damage in femoral artery sections from these groups. Blood levels of glucose (**E**) and body weight (**F**) were also assessed at the end of the experiment, at week 12. Presented *p* values are all significant. * *p* < 0.0001 versus control. LEAD: lower extremity arterial disease; TEM: transmission electron microscopy; T2DM: type 2 diabetes mellitus.

**Figure 2 biomedicines-11-00361-f002:**
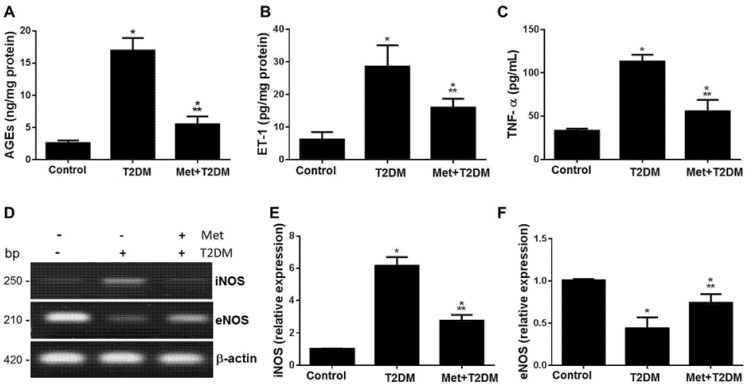
Induction of AGEs, ET−1, and iNOS tissue expression and inflammation secondary to diabetes are inhibited by metformin. Femoral artery tissue levels of AGEs (**A**), ET−1 (**B**), iNOS (**D,E**), and eNOS (**D,F**), as well as blood levels of TNF-α (**C**) were assessed at the end of the experiment in the control group, model group (T2DM), and treated group (Met + T2DM). Presented *p* values are all significant. * *p* ≤ 0.001 versus control, ** *p* < 0.0001 versus T2DM. T2DM: type 2 diabetes mellitus; Met: metformin; AGEs: advanced glycation end products; ET−1: endothelin−1; TNF−α: tumor necrosis factor-alpha; iNOS: inducible nitric oxide synthase; eNOS: endothelial nitric oxide synthase.

**Figure 3 biomedicines-11-00361-f003:**
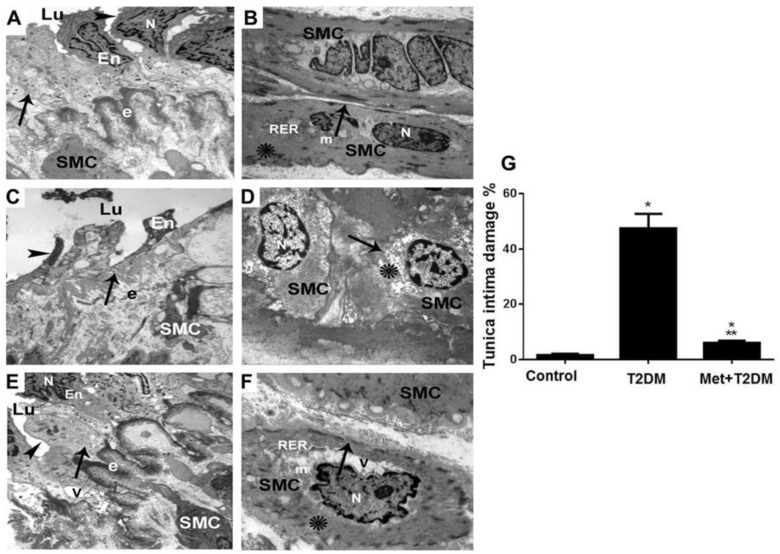
Metformin protects femoral artery ultrastructure against alterations secondary to diabetes. TEM images ((**A**,**C**,**E**) ×6000; (**B**,**D**,**F**) ×10,000) of femoral arteries harvested from rats at the end of the experiment, at week 12 of control untreated group (**A**,**B**), diabetic group (T2DM) (**C**,**D**), and treated group (Met + T2DM) (**E**,**F**) are illustrated. Note that the arrowheads in (**A**,**E**) point to the plasma membrane, and arrowhead in (**B**) points to the disrupted plasma membrane. The arrows in (**A**,**E**) point to the intact intimal surface, and in (**C**) points to the damaged intimal surface. Whereas, arrows in (**B**,**F**) point to the intact plasma membrane, and in (**D**) points to the irregular-shaped plasma membrane. The asterisks in (**D**,**F**) point to the damaged and intact lattice-like networks of actin and myosin filaments, respectively. N: nucleus; Lu: vascular lumen; En: endothelial cell; m: mitochondria; RER: rough endoplasmic reticulum; e: external elastic lamina; V: vacuole; SMC: smooth muscle cell. The histogram in (**G**) represents a quantitative analysis of % tunica intima damage from the groups above. Presented p values are all significant. * *p* ≤ 0.0269 versus control, ** *p* < 0.0001 versus T2DM. T2DM: type 2 diabetes mellitus; Met: metformin.

**Figure 4 biomedicines-11-00361-f004:**
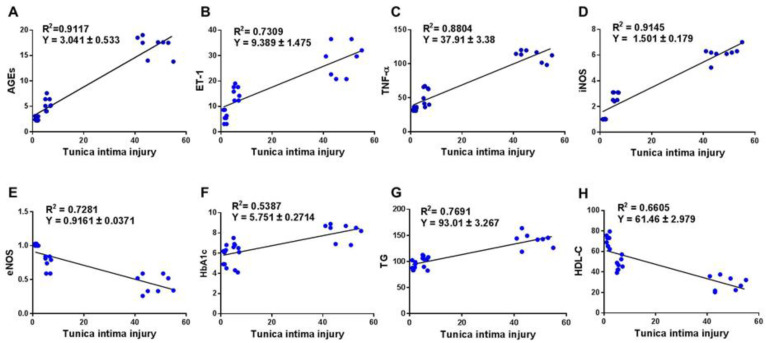
Correlation between the tunica intima injury score and biomarkers of vascular injury and dyslipidemia as well as glycemia. Degree of tunica intima damage was assessed in all harvested femoral arteries at the end of the experiment and the relationship between tunica intima damage versus AGEs (**A**), ET-1 (**B**), TNF-α (**C**), iNOS (**D**), eNOS (**E**), HbA1c (**F**), TG (**G**), and HDL-C (**H**) are shown. AGEs: advanced glycation end products; ET-1: endothelin-1; TNF-α: tumor necrosis factor-alpha; iNOS: inducible nitric oxide synthase; eNOS: endothelial nitric oxide synthase; HbA1c: glycated haemoglobin; TG: triglycerhide; HDL-C: high density lipoprotein-cholesterol.

**Figure 5 biomedicines-11-00361-f005:**
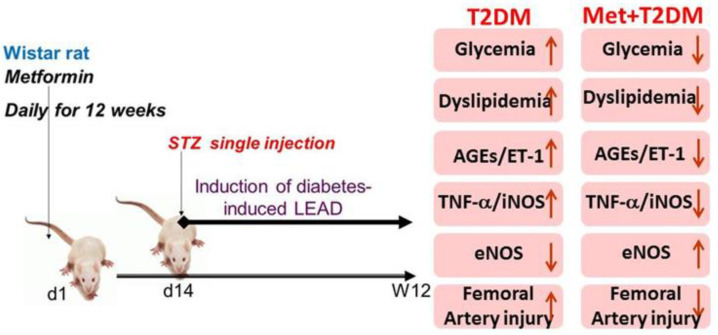
Proposed model for diabetes-induced LEAD appears protected by metformin. LEAD: lower extremity arterial disease; STZ: streptozotocin; T2DM: type 2 diabetes mellitus; Met: metformin; AGEs: advanced glycation end products; ET-1: endothelin-1; TNF-α: tumour necrosis factor-alpha; iNOS: inducible nitric oxide synthase; eNOS: endothelial nitric oxide synthase.

**Table 1 biomedicines-11-00361-t001:** Effects of metformin on diabetes-induced glycemia and dyslipidemia. Blood levels of glucose, HbA1c, TG, CHOL, LDL-C, and HDL-C were assessed 10 weeks following the induction of diabetes in all rat groups. Presented *p* values are all significant (*p* ≤ 0.0369). ^a^: Significant in comparison to control; ^b^: Significant in comparison to T2DM. Met: metformin; HbA1c: glycated hemoglobin; TG: triglyceride; CHOL: cholesterol; LDL-C: low density lipoprotein-cholesterol; HDL-C: high density lipoprotein-cholesterol; T2DM: type 2 diabetes mellitus.

Animal Groups	Glucose (mg/dL)	HbA1c (%)	TG (mg/dL)	CHOL (mg/dL)	LDL-C (mg/dL)	HDL-C (mg/dL)
**Control**	120.3 ± 20.6	5.8 ± 0.9	90.5 ± 6.7	135.0 ± 9.1	46.0 ± 10.97	70.8 ± 6.5
**T2DM**	264.0 ± 32.7 ^a^	8.5 ± 0.3 ^a^	140.1 ± 13.6 ^a^	238.7 ± 21.7 ^a^	182.3 ± 20.56 ^a^	28.5 ± 6.6 ^a^
**Met+T2DM**	158.9 ± 18.0 ^ab^	6.0 ± 1.4 ^b^	104.1 ± 11.4 ^ab^	192.3 ± 9.5 ^ab^	123.8 ± 11.8 ^ab^	47.8 ± 6.3 ^ab^

## Data Availability

The data that support the findings of this study are available on request from the corresponding author.

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
