# Peer review of "Lower Extremity Arterial Disease in Type 2 Diabetes Mellitus: Metformin Inhibits Femoral Artery Ultrastructural Alterations as well as Vascular Tissue Levels of AGEs/ET-1 Axis-Mediated Inflammation and Modulation of Vascular iNOS and eNOS Expression"

_biomedicines, 2023, doi:10.3390/biomedicines11020361_

Round 1

Reviewer 1 Report

I read with great interest the paper “Lower Extremity Arterial Disease in Type 2 Diabetes Mellitus: Metformin Inhibits Femoral Artery Ultrastructural Alterations as well as Vascular Tissue Levels of AGEs/ET-1 Axis-Mediated Inflammation and Modulation of Vascular iNOS and eNOS Expressionby Shati et al.

The design is fine. The article is logically divided into sections and subsections. Statistical analysis is well performed.

Comments:

1.      Introduction, line 49-62: Proinflammatory cytokines (TNF-α, IL-6) are produced by the human adipose tissue, in relation to the degree of obesity. Under physiologic circumstances, adiponectin has anti-inflammatory and anti-fibrotic effects trough suppressing of pro-inflammatory cytokines (TNF-α and IL-6) and inducing anti-inflammatory cytokines (IL-10). In obese patients adiponectin and leptin are impaired, thus promoting an increased inflammatory state (doi: 10.37349/emed.2020.00019).

2.      Discussion: metformin has proven to exert a vascular atheroprotection. The main atheroprotection mechanism depends on insulin-sparing and sensitizing action of metformin that is able to correct the generalized insulin resistance inherent in obesity and T2DM, and the associated systemic dysmetabolic milieu with low-grade inflammation and oxidative stress. The resulting beneficial effects on the arterial wall, both on endothelial and smooth muscle cells, may protect vasculature from fibrosis and remodelling (doi: 10.3390/biom11121834).

3.      Most of the references used are quite old, please add new evidence if available.

4.      Figure 5 has some format problems, please revise.

Author Response

We thank the reviewer for proposing that our manuscript is very interesting and “The design is fine. The article is logically divided into sections and subsections”. We also thank you for your comments which have been taken on board. We have amended the text to highlight these raised points. We hope this is acceptable. Please see the track Changes.

Reviewer 2 Report

This experimental study on a rat model demonstrates that metformin has a protective action on the morphological and pathophysiological damages of diabetes-induced LEAD.

The study is elegant and well written. It is methodologically correct. The conclusions are supported by the results.

This reviewer raises only a few issues that authors need to address.

1- Recently some manuscripts described in detail important extra anti-hyperglycemic effects of metformin, including anti-aging action (Diabetes Research and Clinical Practice, 2020, 160, 108025. doi: 10.1016/j.diabres.2020.108025), and endothelial protection (Biomedicines, 2021, 9 (1), pp. 1-26, 3. doi: 10.3390/biomedicines9010003). Above metformin actions strongly support the results of this study. Therefore, these issues should be addressed by the authors in the discussion.

2- Although this is an experimental study, a section addressing the limitations of the study is missing at the end of the discussion.

3- Figures 1 and 3 contain TEM images in which appear various types of arrows and abbreviations which are not commented in the legends of the figures.

Author Response

We thank the reviewer for proposing that “The study is elegant and well written. It is methodologically correct. The conclusions are supported by the results”. We also thank you for your comments which have been taken on board. We have amended the text to highlight these raised points. We hope this is acceptable. Please see the track Changes.